# Application of CRISPR for In Vivo Mouse Cancer Studies

**DOI:** 10.3390/cancers14205014

**Published:** 2022-10-13

**Authors:** Martin K. Thomsen

**Affiliations:** 1Department of Biomedicine, Aarhus University, 8000 Aarhus, Denmark; mkt@biomed.au.dk; 2Aarhus Institute of Advanced Studies (AIAS), Aarhus University, 8000 Aarhus, Denmark

**Keywords:** CRISPR, mouse models, pre-clinical models, CRISPR screening, CRISPRa, CRISPRi, prime editing, base editing, gene delivery, indel

## Abstract

**Simple Summary:**

Clustered regularly interspaced short palindromic repeats (CRISPR) were discovered in prokaryotes, and the technology can also be used to edit the genome in mammalian cells. The discovery was awarded the Nobel Prize in 2020, as CRISPR has opened up new possibilities to edit the human genome. CRISPR has been applied to study cancer because the method allows for many new ways to model the disease. This includes the development of pre-clinical models of cancer, where CRISPR is used to generate mutations that are found in human cancer. Therefore, unique mutations can be studied in a physiologically relevant setting, and CRISPR technology has accelerated the engineering of these models. This review focuses on exploring the current knowledge of CRISPR editing in adult tissues for generating pre-clinical models to study cancer.

**Abstract:**

Clustered regularly interspaced short palindromic repeats (CRISPR) are widely used in cancer research to edit specific genes and study their functions. This applies both to in vitro and in vivo studies where CRISPR technology has accelerated the generation of specific loss- or gain-of-function mutations. This review focuses on CRISPR for generating in vivo models of cancer by editing somatic cells in specific organs. The delivery of CRISPR/Cas to designated tissues and specific cell compartments is discussed with a focus on different methods and their advantages. One advantage of CRISPR/Cas is the possibility to target multiple genes simultaneously in the same cell and therefore generate complex mutation profiles. This complexity challenges the interpretation of results and different methods to analyze the samples discussed herein. CRISPR-induced tumors are also different from classical tumors in pre-clinical models. Especially the clonal evolution of CRISPR-induced tumors adds new insight into cancer biology. Finally, the review discusses future perspectives for CRISPR technology in pre-clinical models with a focus on in vivo screening, CRISPR activation/inhibition, and the development of prime/ base-editing for the introduction of specific gene editing.

## 1. Introduction

### 1.1. Discovery of CRISPR

Clustered regularly interspaced short palindromic repeats (CRISPR) are multiple short sequences found in the genome of prokaryotes, hence bacteria. These DNA sequences originate from bacteriophages, which had previously infected the bacteria. Hereby, a memory of phage infections is created in the prokaryote and is seen as an adapted immune system by the integration of a unique DNA sequence from the phages into the bacterial genome [1,2]. During a new phage infection, these DNA sequences will be transcribed together with the CRISPR-associated protein (Cas), and by aligning the specific sequences to the genome of the phage, a DNA break can be induced by the Cas protein [3,4,5]. The discovery of this unique immune defense has changed the modeling of many biological processes and was awarded the Nobel Prize in 2020.

### 1.2. The Elements of CRISPR/Cas9

CRISPR has evolved in prokaryotes and resulted in different Cas proteins. Here, Streptococcus pyogenes Cas9 is the most used Cas protein in CRISPR engineering. For genomic DNA to be cleaved by the Cas9 protein, different criteria have to be fulfilled. First, a unique 20-base pair RNA sequence containing the complementary sequence to the genomic target called the guide RNA. This RNA sequence guides the Cas9 protein to the target site and is often fused to a tracrRNA, which binds the Cas9 protein. The whole RNA sequence is called a single guide RNA (sgRNA). Furthermore, a protospacer adjacent motif (PAM) is required for most Cas proteins and, for Cas9 the motive is NGG. The Cas9 protein cleaves the target sequences at position 3+ in the guide sequences and the host organism will repair the break by either homologous or non-homologous end joining (HEJ or NHEJ) [6,7,8]. The homologous repaired genome will maintain the original DNA sequences and can be re-cut by the CRISPR complex. However, after non-homologous end joining, the genomic sequencing is changed by adding or removing base pairs (insertion or deletions (indel)). Hereby, the genomic sequence is changed and can give rise to loss of function mutations in the targeted protein [7,8,9].

The repair through homologous end joining can be used in combination with a “repair template” and hereby a desired genomic sequence can be added to the target of interest. This could be the generation of a driver mutation, such as Kras^G12D^ [10,11]. These are the fundamental principles of CRISPR for genomic editing.

### 1.3. In Vitro Application of CRISPR

After the discovery of CRISPR, the method has been applied to study many biological processes in vitro. As CRISPR has made it possible to genetically edit nearly all types of cells, the method allows for addressing biological questions that were previously difficult or impossible to answer. Different methods are now used to deliver CRISPR/Cas9 in vitro but the most common are plasmid, lentivirus, or electroporation of modified sgRNAs with the Cas9 protein. Each method has its advantage, but it will not be further discussed in this review. The strength of CRISPR is to generate cell lines with genetic alterations that are found in human diseases, including cancer, and the chance to study the implications in vitro. Multiple studies have applied CRISPR to investigate gene functions. Chu et al. (2013) showed as some of the first, that in human embryonic kidney cells *CCR5* can be mutated [8]. Later came Mali et al. (2013), which used CRISPR to insert the coding sequence for GFP in the AAVS1 locus by the use of a repaired template in combination with CRISPR/Cas9 [3]. The specific gene editing by CRISPR with the use of a repaired template is currently optimized by prim- and base-editing and will result in easier and higher efficiency, which is crucial from a therapeutic perspective [12]. Targeting the mitochondrial genome is challenging, as the CRISPR complex is not entering the mitochondrial. Adding a mitochondrial targeting sequence to Cas9 has improved the editing of the mitochondrial DNA and this opens up future treatment options for genetic mitochondrial disorders [13].

The ability of sgRNA and Cas9 to bind specific sequences in the genome has led to the development of CRISPR activation or inhibition (CRISPRa or CRISPRi). Here, a modified Cas9 protein called dead Cas9 (dCas9) is used, as it has lost the ability to cleave the genomic DNA. By fusion of the activating protein VP64 or the repressor protein KRAB to dCas9, these fusion proteins will bind specifically to the promotor region through a sgRNA and alter the expression of the downstream gene [14,15,16,17]. Hereby, it is possible to regulate the expression of a specific gene without the induction of mutations to the genome.

### 1.4. Genome Wide CRISPR Screens

The CRISPR/Cas9 ability to induce mutations with high efficiency has allowed for use of the technology for genome wide screening. These screens have been successful to identify genes that are involved in different malignant processes. Special mutations that gain resistance to drug treatment have been identified through CRISPR screens [18]. To perform CRISPR screens, different considerations have to be taken. Here, a library of sgRNAs targeting the genes of interest has to be generated. The mutation frequency is variable among sgRNAs, therefore multiple guides for the same target genes should be included in the library, and normally 3–5 guides are used for each target [6,19]. Lentiviruses are often used for the delivery of the library, as the virus DNA is integrated into the cell genome. The viral genome can then be used as a bar code to identify which sgRNA has been targeting the cell, and hence, which genes have been mutated. It is also of great importance that each sgRNA is represented equally in the library, so no bias is introduced to the screen. Overall, CRISPR screens have contributed to the discovery of new molecular mechanisms in cancer biology, and work on DNA repair is a strong example where the genetic interaction of PARP has been identified [20], which highlights the importance of CRISPR in cancer research.

### 1.5. In Vivo Application of CRISPR for Cancer Research

After the discovery of CRISPR, the method was fast introduced to generate in vivo models, especially in mice. Now, many new mouse models have been generated by CRISPR by either genetically engineered ES cells or oocyte injections, which is not further discussed in this review [21,22]. Instead, this review focuses on the CRISPR application in somatic cells to generate models for studying cancer. CRISPR technology has been applied to study cancer in vivo by generating loss-of-function or gain-of-function mutations in somatic cells. CRISPRa and CRISPRi have also been used in vivo together with CRISPR screening. The different aspects will be discussed further in this review and their implication for cancer research.

## 2. Delivery of CRISPR/Cas9 In Vivo to Somatic Cells

### 2.1. Delivery by Vector

The transfer of CRISPR editing to in vivo models faces one major hurdle, which is the delivery of the sgRNAs and Cas9 protein to the target cells. This issue is well known in the gene therapy field, and researchers have learned from this field to deliver CRISPR/Cas9 to in vivo models (Figure 1). One of the first studies using CRISPR/Cas9 editing in vivo targeted the liver. As plasmid delivery to the liver can be achieved by intravenous injection, an in vivo expression of sgRNAs and Cas9 protein can be conducted in hepatocytes [23,24]. In that study, liver cancer was induced by CRISPR by mutations to *Trp53* and *Pten*. A plasmid delivery to other cells has very low efficiency but combined with electroporation, it can be increased. By electroporation, gene editing has been done in astrocytes, keratinocytes, muscles, and pancreatic cells [25,26,27,28]. The advantage of plasmid delivery is the ability to deliver a large cargo, as the coding sequence for Cas9 protein is 4.1 kbp. However, the downside is low efficiency, even with electroporation. Electroporation causes tissue damage and is more efficient at the organ developmental stage and not in adult tissues [29]. Therefore, plasmid delivery is nearly limited to target hepatocytes.

### 2.2. Viral Delivery of CRISPR/Cas9

For the delivery of CRISPR/Cas9 to different tissues, viral delivery has advantages, as viruses have tropism for many cell types. Different viruses are used, and the first was lentivirus, as a large cargo can be cloned to the viral genome [30]. This allows for the delivery of the genomic code for sgRNAs and the Cas9 protein to the target cell. Furthermore, the genome of the lentivirus is integrated into the host genome. Hereby, Cas9 and sgRNA will be expressed continually in the transduced cell. This ensures high efficiency of the CRISPR-induced mutation at the target site, but concerns about off-target mutations have been raised, as guides are kept being expressed in the cells together with Cas9 [9,31]. A lentivirus was first applied to induce lung cancer but has later been used in multiple organs. This includes pancreas breast, and brain [32,33,34,35]. The viral integration provides a “fingerprint” in the transformed cell and makes it possible to confirm the sgRNA sequence. However, the integration into the host genome can potentially promote cancer initiation, which is a problem with the use of lentivirus delivery [36].

To avoid the integration of the viral genome into the target cells, adenoviruses and adeno-associated viruses are used for the delivery of CRISPR/Cas9. Both viruses are present transient in the transduced cell and will not be replicated in the host cell during cell division. Adenoviruses can carry large cargo and have tropism to a wide range of cells [37]. However, adenovirus is highly pathogenic and can induce cell death and tissue damage to the target tissues [37]. Adenoviruses are routinely used for the delivery of sgRNAs and Cas9 to induce cell transformation in different tissues in vivo. Again, this delivery method was pioneered in the liver to induce hepatocellular carcinoma (HCC) but has been used intensely to induce lung and brain cancer in mice [38].

An adeno-associated virus (AAV) is different from AV, as biologically it can only be reproduced in cells that are co-infected with AV [39]. AAV is not pathogenic and can be present in the infected cell for more than a year. It is rarely integrated into the host genome and the integration is nearly always at a specific site, which is not interfering with the expression of other genes [39]. Different serotypes of AAV have strong trophy for specific cell types, and this can improve the transduction and induction of cancer at designated sites [40]. One problem with the use of AAV is the cargo site. The virus particles can hold up to approximately 5 kilobases, and as the coding sequence for Cas9 is 4.1 kilobases, this reduces the options for the delivery of multiple sgRNAs together with Cas9. Small variants of Cas9 have thus been engineered to overcome this problem [41]. As an alternative, multiple mouse strains transgenic for the Cas9 protein have been developed. The use of these strains has multiple advantages when combined with the delivery of sgRNAs (Figure 2). In the first study that uses transgenic Cas9 mice, models of lung, brain, and skin cancer were generated by the delivery of AAV particles containing multiple sgRNAs. Furthermore, the expression of Cas9 in this mouse strain is conditional, and Cre protein has to be present in the cells to recombine away from three-stop codons flanked by two loxP sites. Therefore, Cre expression in this study was delivered by the AAV construct together with the sgRNA [10]. As an alternative, the transgenic Cas9 stain could be bred to a tissue-specific Cre line, which would ensure that Cas9 expression is restricted to the cells of interest. The use of Cas9 transgene mice together with the viral delivery of sgRNA has biosafety advantages. By making it a two-combination system, self-inoculation with viral particles will have fewer implications and ease work procedures.

### 2.3. Lipid Nanoparticle for In Vivo Delivery of CRISPR/Cas9

Alternative methods for the delivery of CRISPR guides and the Cas9 protein are continually under development. In vitro application of modified sgRNAs in complex with Cas9 protein (Cas9/sgRNA ribonucleoprotein complexes (RNPs)) is delivered efficiently by electroporation [43]. However, for an in vivo application, RNPs are not efficient, as they degrade rapidly and are negatively charged, which compromises the uptake [44]. Therefore, lipid nanoparticles are used to pack RNPs for delivery to different tissues. Successful gene editing has been conducted in different tissues, such as the liver, brain, and lung to induce cancer. Some lipid nanoparticles even have preferences for different organs, but the induction of gene editing in non-desired cells or tissues could be a problem with this method [44].

The development of RNA vaccines has opened the possibility to deliver mRNA for Cas9 and sgRNA to the target cells to let Cas9 protein and sgRNA be synthesized in the cell. The production of nano-particles can accelerate the speed and reduce the cost of applying in vivo editing by CRISPR/Cas. This will result in faster models at reduced costs and benefit the research into cancer-causing mutations.

## 3. CRISPR Induced Tumors

### 3.1. Targeting Multiple Genes Simultaneously for Cancer Induction

Human tumors contain multiple mutations and modeling this high complexity in vivo has been limited by interbreeding multiple mouse strains. One of the advantages of a CRISPR application in vivo is the delivery of sgRNAs to multiple targets in the same cell. Hereby, it is possible to edit several tumor suppressor genes simultaneously and investigate the potential cross-talk in tumor initiation and progression. Many groups have delivered 3–4 sgRNAs, which has accelerated tumor formation [10,45]. We have delivered up to 8 sgRNAs simultaneously with success (unpublished work), and the limitation is at the cargo site of the viral vector. Analysis of mutation profiles of the tumors clearly indicates that multiple mutations of tumor suppressor genes accelerate tumor progression [46]. Strong tumor suppressor genes such as Pten are always found mutated in prostate cancer, whereas less essential mutations can be found intact in a subset of tumors [45,47]. Similarly, gain-of-function mutations engineered by homologous repair have low efficiency in vivo [17,48]. It has been shown that only a few percent of the mouse lung tissues had Kras^G12D^ mutations [10], but an analysis of the tumors revealed that these mutations were found in a quarter of the samples [46]. The heterogeneity of the CRISPR-induced mutation profile provides a base to follow and investigate tumor evaluation, as the clones with the highest fitness will overgrow less aggressive ones. To decrease the heterogeneity induced by the delivery of multiple sgRNAs, CRISPR editing has been combined with traditional mouse strains with loss of *Brca1* and *Trp53*. Annunziato et al., (2019) investigated the loss of either *Pten* or *Rb1* by CRISPR in the breast tissues of mice with germline of tissue-specific loss of *Brca1* and *Trp53.* By this experimental design, the deficiency of Brca1 and Trp53 was constant and the implication of *Pten* or *Rb1* could be delineated [29].

### 3.2. Insertion and Deletion Analysis for Tumor Profiling

The indel profile for the target site is commonly analyzed from amplified PCR products and Sanger sequences. Different web tools can analyze the indel profile and provide important information about the tumor context. As the data will show, a percent of the sample maintains the wild-type sequences. Bystander cells in the tumor microenvironment will not be mutated, and the indel analysis gives an estimate for the percent of tumor cells. However, this analysis has pitfalls, as cells with heterogenic mutations could be present. Here, an analysis of multiple target genes will provide evidence for potential heterozygote mutations. The indel analysis will also indicate if the tumor is clonal or contains multiple subclones (Figure 3). As tumor cells can duplicate the DNA and change the chromosomal number, this analysis cannot stand alone. One large advantage is the “fingerprint” of the tumor, which is provided by the indel profile. It is possible to identify metastasis and map them with the primary tumor. It is unlikely that the same indel profile occurs for multiple target genes in different tumors [23]. We identified a metastasis in the abdomen of a mouse, which contained identical indels profile for 3 target genes as the primary lung tumor [46]. This shows that genetic editing by CRISPR/Cas9 can be used for in-depth analysis of tumor biology.

### 3.3. Chromosomal Rearrangement by CRISPR In Vivo

Chromosomal rearrangement is a hallmark of oncogenesis and is found in different tumors. These fusion genes have been studied in transgenic mice by overexpression of the fusion gene [49]. These fusion genes can now be generated by CRISPR. In lung cancer, the fusion between ELM4 and ALK occurs in more than 5% of non-small-cell lung cancers and is a known driver mutation. By targeting the intron in these two genes by sgRNA, Blasco et al., (2014) recapitulated this subtype of lung cancer in a mouse [50]. Other fusion genes have been generated and a rear fusion of NTRK1 and BCAN, which drive tumor development in different tissues, was shown to induce glioma [51]. Generation of fusion mutations in vivo has the advantage of clonal selection if the fusion gene provides a growth advantage. Hereby the fusion product is seen in the majority of tumor samples [51]. Overall, mouse models of rear gene fusion are important, as pre-clinical models can be used to study treatment response and resistance mechanism [51].

### 3.4. In Vivo CRISPR Screen

One advantage of CRISPR/Cas9 is the possibility to screen a large set of genes. In vitro, a number of genome wide screens have been performed to great success. The advantage of in vitro screening is the large number of cells that can be used, which ensures coverage of the entire library. Libraries with 50.000 to 100.000 unique sgRNAs can be screened efficiently. However, these libraries are difficult to apply to in vivo studies, as each sgRNA should be presented a number of times. Even if 100 tumors can be induced in the organ of interest, many mice have to be included in the study, and the downstream analysis of the tumors is very laborious. Therefore, in vivo screens have been targeted to a subgroup of genes. The work by Chow and Wang (2017 & 2018) generated a library containing more than 250 tumor suppressor genes represented by 5 different sgRNAs cloned into an AAV vector. Furthermore, to ensure tumor formation in a manner of time, the construct contained a sgRNA for *Pten*, and the screen was performed in mice deficient for *Tpr53.* Hereby, 3 tumor suppressor genes were targeted, and this library was used to investigate glioma and HCC formation in mice [52,53].

CRISPR activation or inhibition screens in vivo are still to be published. Transgenic mice with dCas9 protein fused to KRAB for inhibition or VP64 and other variations for CRISPR activation have been generated [54,55,56,57]. In addition, lentivirus libraries containing a guide to the promoter regions have been conducted and are available at Addgene. The tools to perform in vivo screenings are in place but the complexity of analyzing samples and the identification of genes that have been altered by CRISPR is labor intensive as seen for traditional CRISPR screens in vivo [58]. However, in vivo CRISPR activation or inhibition screens will be very informative, as many gene functions are different in an in vivo setting, and this will increase the understanding of cancer biology.

## 4. Pit Falls by CRISPR/Cas9 in In Vivo Cancer Modeling

### 4.1. Tumor Heterogeneity

CRISPR/Cas-induced tumors are different from traditional mouse models of cancer. As traditional mouse models have germline mutations or a conditional allele to induce tissue-specific mutations, the entire organ holds the same mutation. In contrast, CRISPR-induced mutation in somatic cells will only occur in a few cells and give rise to clonal expansion, as seen in human cancers [45]. One issue with inducing cancer through CRISPR is the efficiency of the sgRNA when mutating multiple genes simultaneously. This generates clones with different mutation profiles, as some genes will not be edited by CRISPR [47]. However, this is also an advantage, as clones with different profiles will be generated and cancer Darwinian evolution will be subjected to the clones [59]. This imperfection of CRISPR is generating different tumor mutation profiles, which can serve as controls and be used for the interpretation of data [46]. Nevertheless, for the researcher, each tumor mutation burden has to be assessed, and tumors should be sub-grouped before being analyzed.

### 4.2. Biased Indel Analysis and off Target Mutations

The base editing by NHEJ of CRISPR-induced mutations is random and cannot be controlled. Most of the indels will lead to changes in the reading frame and to an introduction of a pre-mature STOP codon. However, indels of 3 bases will not disrupt the reading frame but only remove or add an amino acid. To increase the possibility for loss of function mutations and not a truncated protein, the CRISPR guides should be designed to the 5′ end of the gene. Sanger sequencing in combination with web tools for analysis provides the indel profile and the KO efficiency. The indel analysis relies on PCR amplification of the breakpoint, and here large indels or even chromosomal re-arrangement can be undetected. Therefore, the indel assessment could be complemented with a protein analysis by Western blot or immunohistochemistry, to confirm loss-of-function mutations [45,46]. As the analysis of the PCR fragment has limitations, whole genome sequencing (WGS) of the tumor will identify all genomic alterations. WGS is also powerful to assess off-target mutations generated by the sgRNA. Web tools such as CRISPRO for sgRNA design provide predictions for off-target mutations. The implication of off-target mutations with the application of CRISPR editing is widely discussed, but the implication on in vivo modeling of cancer is probably minor [60]. Very few off-target mutations have been identified and have not been linked to a tumor-promoting function. Unpublished work by us has identified an off-target mutation in an intronic region. However, we could not reveal any functional implication of this particular mutation and therefore classified it as a “passenger mutation”. It is very unlikely to see the same off-target mutation in multiple tumors, and these mutations add to the heterogeneity of the tumor, as seen in humans.

### 4.3. Delivery and Non-Intended Tumors Formation

A bottleneck for use of CRISPR/Cas9 to model cancer in vivo is the delivery of the sgRNA and Cas9 protein. It limits which organs can be targeted, and especially the crypt structure in the colon proves difficult. We have delivered AAV to the prostatic lobes of the mouse, where a surgical procedure followed by a complex injection is performed [61]. Under these circumstances, the delivery can fail and no tumor initiation will occur in the targeted organ. However, improper injection of the viral particles can lead to tumors in other organs and this is also seen with the delivery of nanoparticles. Therefore, tumor formation that occurs in other organs should be thoroughly profiled to reveal if these are metastases or primary tumors. Here, the indel profile of the primary tumor should be compared to the potential metastasis, and this analysis should be followed by a pathology examination [46]. To overcome the problem of tumor formation in secondary tissues, an expression of Cas9 can be tissue specific. By the use of transgene mice for Cas9 under Cre induction, tissue-specific Cre expression can be applied. Tissue-specific mouse strains for Cre expression can be intercrossed with conditional Cas9-expressing mice. As an alternative, Cre expression by a tissue-specific promoter can be cloned to a viral backbone and delivered in the viral particle.

## 5. Conclusions

The discovery of CRISPR/Cas9 has changed the way to model human cancer in vitro and in vivo. The application of CRISPR/Cas9 to study cancer in vivo will continue to develop and be applied to different species, not restricted to the mouse [28,62,63]. CRISPR/Cas9 has mainly been used to study loss-of-function mutations of potential tumor suppressor genes. However, the development of base- and prime-editing by CRISPR allows for modeling gain-of-function without the use of classical homologous repair templates [64]. Special prime editing seems efficient and will be further optimized so that it can be applied to in vivo cancer studies.

Another crucial aspect of in vivo modeling by CRISPR/Cas9 is the delivery. Here, the rapid development of nanoparticles for the delivery of RNP or mRNA will complement the viral delivery platform. The special multiplex of sgRNA in somatic cells is a powerful property of the CRISPR/Cas9 technology compared to classical mouse models [65]. These methods will accelerate the modeling, and as WGS has become more accessible, this method can be applied to the downstream analysis, which only will become more complex as the complexity of the tumors are increasing. Overall, CRISPR technology has changed the modeling of cancer, and especially the in vivo application of it will bring a new understanding to cancer biology and help to develop new treatments.

## Figures and Tables

**Figure 1 cancers-14-05014-f001:**
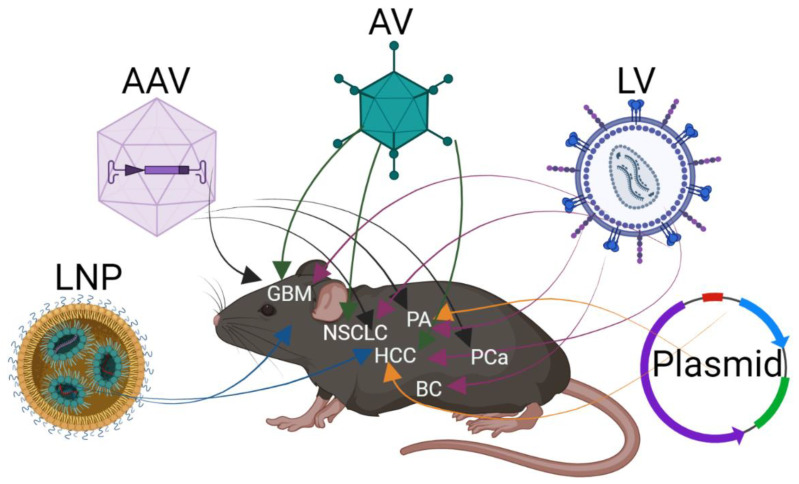
Delivery of CRISPR/Cas to induce cancer in the mouse. Schematic overview of delivery of sgRNA and Cas protein to different mouse organs. Plasmid delivery has mainly been done to the liver but also to astrocytes, keratinocytes mussel and pancreatic cells by either intravenous injections or by electroporation. Lentivirus (LV) has been applied to deliver CRISPR to the liver, pancreas, breast, lung and brain cancer. Adeno virus (AV) has been used to induce lung, liver, and glioma cancers. Adeno associated virus (AAV) is used to induce lung, liver, glioma, and prostate cancers by delivery through injections, operation or inhaling. Lipo nano particles (LNP) have emerged for CRISPR/Cas delivery and have been applied to GBM, liver and lung cancer.

**Figure 2 cancers-14-05014-f002:**
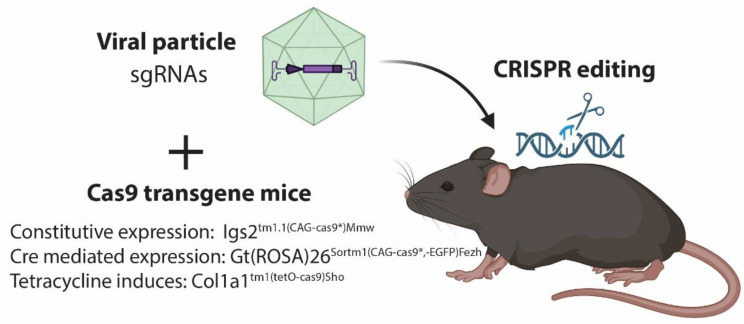
Two combination system for in vivo CRISPR editing. Combination of viral particles with transgenic mice for Cas9 expression has multiple advantages. Cas9 can be ubiquitously expressed [34] or controlled by Cre expression [10] or tetracycline [42]. Cas9 protein is not delivered by viral particles and does not take space from the viral genome. Furthermore, by separating Cas9 and sgRNA expression, biosafety is increased as self-inoculation will not contain a risk for CRISPR editing of the scientist.

**Figure 3 cancers-14-05014-f003:**
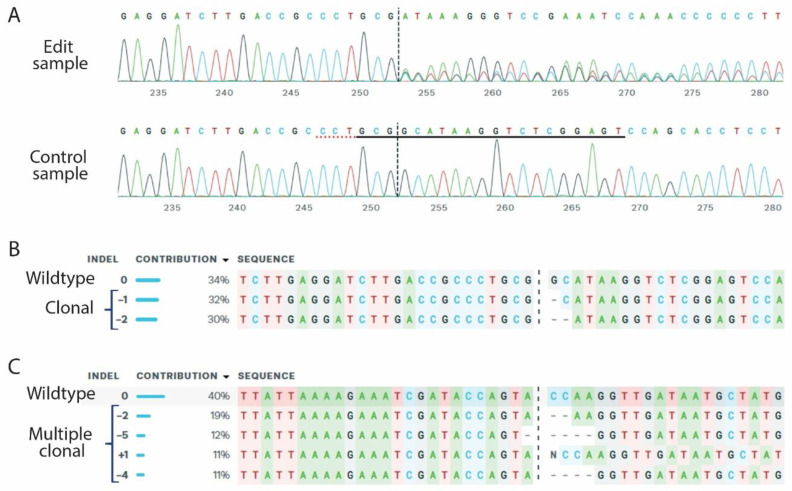
Indel analysis of CRISPR induced tumors. (**A**) Base call from and edit samples and a control at the target site of the sgRNA (marked with a black line and PAM site with a red dash line). Preferable editing site is a 3+ of the sgRNA (black dash line). (**B**) Indel analysis of a tumor, which predicted to be clonal. The analysis shows that 34% of the genome is unedited. 32% of the target has 1 base pairs deletion and 30% has 2 base pairs deletion. This suggests that ~60% of the cells are edited at the target site and ~40% of the cells not edited and could be cells of the tumor microenvironment. (**C**) Indel analysis of a tumor, which predicted to be polyclonal. The analysis shows that 40% of the genome is unedited. Of the edit genome, different indel’s on 19% (−2 base parries), 12% (−5 base parries), 11% (+1 base pare) and 11% (−4 base parries). This suggests that the tumor contains two clones, or that genomic amplification has taken place.

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
