# Peer review of "Application of CRISPR for In Vivo Mouse Cancer Studies"

_cancers, 2022, doi:10.3390/cancers14205014_

Round 1

Reviewer 1 Report

This review written by Thomsen summarized the cutting-edge science of using CRISPR to generate pre-clinical models for cancer research, covered most of the main advances and also gave very insightful views on the pitfalls and future perspectives. The technology of CRISPR-mediated cancer model generation is of great significance that is bringing evolution to cancer research and treatment. the cancer-modelling field is rapidly growing, so I reckon this timely review will be necessary as an update. Therefore, I recommend this review be published on Cancers. The following suggestions can help to improve the scientific contributions of this review:

Some statements are very general, adding more supportive evidence to the statements will help readers' understanding and improve the content of this review. For example, the authors can present and discuss a few related and important advances to illustrate how CRISPER screens facilitate the discovery of molecular mechanisms in cancer biology in the statement "CRISPR screens have contributed to the discovery of new molecular mechanisms in cancer biology, and this highlights the importance of CRISPR in cancer research". This suggestion is also applied to other general statements across the review.

References needed for the following statements:

“The CRISPR technology has been applied to study cancer in vivo by generating loss-of-function or gain-of-function mutations in somatic cells”;

“Electroporation causes tissue damage and is more efficient at the organ developmental stage and not in adult tissues”;

“In vitro application of modified sgRNAs in complex with Cas9 protein (Cas9/sgRNA ribonucleoprotein complexes (RNPs)) is delivered efficiently by electroporation”;

Please double-check and make sure every conclusion drawn from existing studies has supporting references placed. 

Ref.19 and ref.20 references are a little outdated, could the authors replace them with some up-to-date reputable ones so the readers are able to keep up with the latest developments by reading this review?

“HCC” should be explained and not be abbreviated on the first use.

Create a figure to communicate the complex information in the “two-combination system”. Also, ref 10 is a key study of the cancer-modelling field, it is worth having a scheme to describe this work. 

In section 3, targeting multiple genes simultaneously for cancer induction was reviewed, please also brief review the history of cancer model induction with only one gene engineered, discuss the advantages of engineering multiple genes over engineering only one gene in inducing tumours as well as the disadvantages. Which mode is more biologically relevant (closer to the wild-type tumour context)?

Describe the mechanism of CRISPR-mediated chromosomal rearrangement in section 3.3. This application is less familiar for the researches especially who are not specialized in this field.

“to ensure tumor formation in a manner of time, 266 the construct contained a sgRNA for Pten”: “Pten” gene names should be italicized.

Little messages are delivered in the paragraph from line 270 to 277. Elaborate the obstacles that impede CRISPR activation or inhibition screens in vivo. Compare the in vivo screening by CRISPR-mediated gene editing with that by CRISPRi/CRISPRa.

Given the main focus of this review is to explore the current knowledge on CRISPR editing in adult tissues for generating pre-clinical models to study cancer, I suggest making a table/figure/scheme/timeline to list the key developments in this field. This table/figure/scheme/timeline can include but not limited to the cancer type/tissue model type, engineered genes, types of gene engineering (e.g. mutations/deletions/indels), consequences of gene engineering (e.g. loss-of-function/gain-of-function), CRISPR/Cas type, CRISPR/Cas delivery mode, therapeutics, clinical state (if entered clinic trial), relevant publications.

Author Response

Point-by-point letter

I thank the reviewers for the constructive and positive revision. I have applied the majority of the suggestions to the review and find it has improved the manuscript overall. Note, a new figure 2 has been added illustrating use of Cas9 transgene mice and viral particles for sgRNA delivery.

This review written by Thomsen summarized the cutting-edge science of using CRISPR to generate pre-clinical models for cancer research, covered most of the main advances and also gave very insightful views on the pitfalls and future perspectives. The technology of CRISPR-mediated cancer model generation is of great significance that is bringing evolution to cancer research and treatment. the cancer-modelling field is rapidly growing, so I reckon this timely review will be necessary as an update. Therefore, I recommend this review be published on Cancers. The following suggestions can help to improve the scientific contributions of this review:

Some statements are very general, adding more supportive evidence to the statements will help readers' understanding and improve the content of this review. For example, the authors can present and discuss a few related and important advances to illustrate how CRISPER screens facilitate the discovery of molecular mechanisms in cancer biology in the statement "CRISPR screens have contributed to the discovery of new molecular mechanisms in cancer biology, and this highlights the importance of CRISPR in cancer research". This suggestion is also applied to other general statements across the review.

  • This good suggestion has been added to different places in the review.

References needed for the following statements:

“The CRISPR technology has been applied to study cancer in vivo by generating loss-of-function or gain-of-function mutations in somatic cells”;

  • This is an intro sentence to the next paragraph and is discussed later in the review. Therefore, this is not elaborated in section 1.5.

“Electroporation causes tissue damage and is more efficient at the organ developmental stage and not in adult tissues”;

  • A reference reviewing the plasmid delivery by electroporation has been added to the manuscript.

“In vitro application of modified sgRNAs in complex with Cas9 protein (Cas9/sgRNA ribonucleoprotein complexes (RNPs)) is delivered efficiently by electroporation”;

  • A reference on modified sgRNA in RNP complexes has been added for the in vitro work.

Please double-check and make sure every conclusion drawn from existing studies has supporting references placed. 

  • This has been done.

Ref.19 and ref.20 references are a little outdated, could the authors replace them with some up-to-date reputable ones so the readers are able to keep up with the latest developments by reading this review?

  • It is true that both references are old but both works have been pioneer in the field. As this review is not focusing on the in vitro or germline CRISPR editing, I have chosen to keep the original articles and not the news advances in those areas.   

“HCC” should be explained and not be abbreviated on the first use.

  • This has now been corrected.

Create a figure to communicate the complex information in the “two-combination system”. Also, ref 10 is a key study of the cancer-modelling field, it is worth having a scheme to describe this work. 

  • This good suggestion has been applied and a new figure 2 is showing the two-combination system and alternative Cas9 transgene mice.

In section 3, targeting multiple genes simultaneously for cancer induction was reviewed, please also brief review the history of cancer model induction with only one gene engineered, discuss the advantages of engineering multiple genes over engineering only one gene in inducing tumours as well as the disadvantages. Which mode is more biologically relevant (closer to the wild-type tumour context)?

  • I thank the reviewer for the valuable comment. I have now added the work by Jonkers group, where loss of Pten or RB1 has been combined with traditional mouse models to keep the modeling more simple and strengthen the data. This is on line 225-230.

Describe the mechanism of CRISPR-mediated chromosomal rearrangement in section 3.3. This application is less familiar for the researches especially who are not specialized in this field.

  • Line 263 contains a sentence for generation of fusion genes: By targeting the intron in these two genes by sgRNA, Blasco et al., (2014) recapitulated this subtype of lung cancer in a mouse.

Furthermore, it has now been added that fusion genes can undergo positive selection in in vivo studies, which can promote detection of these rare fusion. See line 263-269.    

“to ensure tumor formation in a manner of time, 266 the construct contained a sgRNA for Pten”: “Pten” gene names should be italicized.

  • This mistake has been corrected.

Little messages are delivered in the paragraph from line 270 to 277. Elaborate the obstacles that impede CRISPR activation or inhibition screens in vivo. Compare the in vivo screening by CRISPR-mediated gene editing with that by CRISPRi/CRISPRa.

  • It has been added that the same complexity applies to activation and inhibition screens as for traditional in vivo CRISPR screening and a new reference on in vivo screening has been added. See line 291-293.

Given the main focus of this review is to explore the current knowledge on CRISPR editing in adult tissues for generating pre-clinical models to study cancer, I suggest making a table/figure/scheme/timeline to list the key developments in this field. This table/figure/scheme/timeline can include but not limited to the cancer type/tissue model type, engineered genes, types of gene engineering (e.g. mutations/deletions/indels), consequences of gene engineering (e.g. loss-of-function/gain-of-function), CRISPR/Cas type, CRISPR/Cas delivery mode, therapeutics, clinical state (if entered clinic trial), relevant publications.

  • This point has partly been addressed by adding the new figure 2.

Reviewer 2 Report

The paper "Application of CRISPR for in vivo cancer studies" is a well thought and organized review reflecting the current state of CRISPR technology application in the cancer research field.  The manuscript largely covers the latest knowledge of CRISPR usage in somatic cells for cancer research, primarily in vivo and some in vitro. Genome screen, delivery systems or tumor incident are also well summarized, with reference to some of the genes that have been targeted with CRISPR to induce tumor onset. Advantages and pitfalls are too addressed,

However, I would like to know which are the latest development in targeting nuclear-encoded mitochondrial genes with CRISPR and their contribution to cancer initiation. Additionally, if there's been enough evolution to target mitochondrial DNA with CRISPR technology.  I think that this information would complement and strengthen the manuscript. 

Author Response

Point-by-point letter

I thank the reviewers for the constructive and positive revision. I have applied the majority of the suggestions to the review and find it has improved the manuscript overall. Note, a new figure 2 has been added illustrating use of Cas9 transgene mice and viral particles for sgRNA delivery.

The paper "Application of CRISPR for in vivo cancer studies" is a well thought and organized review reflecting the current state of CRISPR technology application in the cancer research field.  The manuscript largely covers the latest knowledge of CRISPR usage in somatic cells for cancer research, primarily in vivo and some in vitro. Genome screen, delivery systems or tumor incident are also well summarized, with reference to some of the genes that have been targeted with CRISPR to induce tumor onset. Advantages and pitfalls are too addressed,

However, I would like to know which are the latest development in targeting nuclear-encoded mitochondrial genes with CRISPR and their contribution to cancer initiation. Additionally, if there's been enough evolution to target mitochondrial DNA with CRISPR technology.  I think that this information would complement and strengthen the manuscript. 

I thank the reviewer for this comment. To my knowledge, in vivo editing of mitochondrial DNA has not been published. Instead it has been done in vitro and the method is currently optimized by delivery of Cas9 and sgRNA to the mitochondrial. This has been added to the manuscript on line 80-83.  

Round 2

Reviewer 1 Report

suggest acceptance of this review paper